# Artificial Intelligence in the Non-Invasive Detection of Melanoma

**DOI:** 10.3390/life14121602

**Published:** 2024-12-04

**Authors:** Banu İsmail Mendi, Kivanc Kose, Lauren Fleshner, Richard Adam, Bijan Safai, Banu Farabi, Mehmet Fatih Atak

**Affiliations:** 1Department of Dermatology, Niğde Ömer Halisdemir University, Niğde 51000, Turkey; 2Dermatology Service, Department of Medicine, Memorial Sloan Kettering Cancer Center, New York, NY 10021, USA; kosek@mskcc.org; 3School of Medicine, New York Medical College, Valhalla, NY 10595, USA; lfleshne@student.touro.edu (L.F.); radam2@student.touro.edu (R.A.); bijan_safai@nymc.edu (B.S.); banufarabi91@gmail.com (B.F.); 4Dermatology Department, NYC Health + Hospital/Metropolitan, New York, NY 10029, USA; fatih9164@hotmail.com; 5Dermatology Department, NYC Health + Hospital/South Brooklyn, Brooklyn, NY 11235, USA

**Keywords:** artificial intelligence, algorithms, melanoma, skin cancer, dermoscopy, non-invasive skin imaging, reflectance confocal microscopy, optical coherence tomography, diagnostic accuracy, skin cancer detection

## Abstract

Skin cancer is one of the most prevalent cancers worldwide, with increasing incidence. Skin cancer is typically classified as melanoma or non-melanoma skin cancer. Although melanoma is less common than basal or squamous cell carcinomas, it is the deadliest form of cancer, with nearly 8300 Americans expected to die from it each year. Biopsies are currently the gold standard in diagnosing melanoma; however, they can be invasive, expensive, and inaccessible to lower-income individuals. Currently, suspicious lesions are triaged with image-based technologies, such as dermoscopy and confocal microscopy. While these techniques are useful, there is wide inter-user variability and minimal training for dermatology residents on how to properly use these devices. The use of artificial intelligence (AI)-based technologies in dermatology has emerged in recent years to assist in the diagnosis of melanoma that may be more accessible to all patients and more accurate than current methods of screening. This review explores the current status of the application of AI-based algorithms in the detection of melanoma, underscoring its potential to aid dermatologists in clinical practice. We specifically focus on AI application in clinical imaging, dermoscopic evaluation, algorithms that can distinguish melanoma from non-melanoma skin cancers, and in vivo skin imaging devices.

## 1. Introduction

Skin cancer is the most commonly diagnosed cancer among fair-skinned populations with an increasing incidence worldwide [1]. Cancers of the skin are typically defined as either melanoma or non-melanoma. Melanoma, the most lethal among skin cancer subtypes, occurs due to the uncontrolled proliferation of melanocytes [2]. The American Cancer Society reports that although melanoma cases constitute only 1% of total skin cancer cases, death rates from melanoma are much higher compared to other skin cancer subtypes [3].

An early diagnosis of skin cancer, especially melanoma, is highly effective in reducing mortality [4]. Currently, skin biopsies and histopathological evaluation are the gold standard in the diagnosis of skin cancer [5]. However, confirming all skin lesions with a biopsy is impractical for several reasons, including scar formation from excisions, time constraints in clinical practice, and financial burdens. As a result, several imaging technologies are utilized to determine the necessity of a biopsy [6,7]. One example is dermoscopy, an epiluminescence microscopy technique that utilizes a magnifying lens and a (non-)polarized light source to capture subsurface morphologic features (including pigmentation) from epidermal and dermal layers of the skin. Dermoscopy is also widely used in the diagnosis of skin diseases, especially skin cancers. Furthermore, the use of high-resolution, non-invasive diagnostic devices such as confocal microscopes, which can acquire images of skin lesions at the cellular resolution, on par with histology, has also become widespread [8,9].

The use of such imaging technologies is successful in reducing unnecessary biopsies and increasing sensitivity; however, their success is highly correlated to the skill level of providers. For instance, while many residency programs incorporate imaging techniques into teaching, no such training exists in dermatology residency programs. Therefore, the clinicians’ performance in utilizing these technologies for diagnostic assessment is variable and highly user-dependent.

Recently, significant strides have been made to streamline the diagnosis of skin cancer and provide more rapid diagnoses, such as in primary healthcare settings, with the utilization of AI. AI algorithms have been designed to incorporate macroscopic, dermoscopic, and histopathological images to predict suspicious lesions that warrant further testing. Prior literature has demonstrated that AI algorithms can perform as well as or better than consultant dermatologists and can assist clinicians in the diagnosis of skin cancers [10,11].

In this review, we aim to discuss the current status of utilizing AI-based technology in the non-invasive diagnosis of melanoma, their potential applications, and their drawbacks.

### 1.1. Artificial Intelligence: Fundamental Principles

Artificial intelligence encompasses a wide spectrum of technologies that empower machines to simulate human-like intelligence, problem-solving abilities, and cognitive functions. Machine learning (ML) is a subfield of AI that can make predictions based on user input data [12]. ML presents an excellent opportunity for the automation of medical data analysis to impact clinical care, with its ability to learn and make predictions that can potentially support clinical decision-making processes.

Supervised models are currently the most prevalent form of ML utilized in dermatology. In this approach, each sample in the dataset is associated with “label(s)”. During the training process, the model learns to estimate labels from the raw data of the samples, such as pixel values for images. The three primary tasks undertaken are classification, detection, and segmentation. In classification, each sample is associated with a label, such as a dermoscopy image classified as melanoma. Detection involves identifying the presence or absence of a given structure within the sample, such as detecting atypical networks in a dermoscopy image. Segmentation moves a step further by identifying the existence by location and delineating the extent of the structure by outlining its borders in the image. These exemplar applications of skin cancer detection with AI algorithms are shown in Figure 1.

Currently, the majority of the ML studies in dermatology involve applications of deep learning (DL) models (e.g., convolutional neural networks (CNN), transformers, or their variants/combinations) to classify images to improve the diagnosis of skin diseases [13]. Unlike traditional machine learning (ML) approaches that rely on hand-crafted features extracted from dermatology images to capture human-interpretable characteristics such as texture, color, and border information, the deep learning (DL) models leverage more sophisticated feature extraction techniques that can uncover complex correlations within the data samples to optimize target success measures like classification accuracy, detection precision, or segmentation performance. By using learnable features driven by optimization processes tied to the success metric, DL models can extract higher-order representations from the data that are not easily discernible through traditional methods.

In their most basic and widely used form, CNNs consist of multiple cascading non-linear modeling units called “layers”. These layers filter the input data by filtering the redundant information, finding correlations, and summarizing critical information into a distilled representation called “features”. These layers are typically followed by several classification layers, which map the extracted features to target diagnostic labels. In more recent ML models, CNNs have been largely superseded by newer architectures, namely Transformers, which leverage the ability to capture long-range dependencies and contextual information within sequential data such as text. This is achieved through a mechanism called “attention”, where the model selectively focuses on relevant parts of the input data to generate output. Unlike CNNs, which possess limited contextual information extraction capabilities, Transformers excel at tasks requiring the capture of long-range contextual information in the data, leading to widespread adoption across various domains. Furthermore, Transformer models have been adapted for visual tasks like image classification and segmentation, with Vision Transformers being a notable variant. In this context, Vision Transformers process images by dividing them into smaller patches (analogous to words) and encoding them through self-attention mechanisms to discover global relationships between them. Vision Transformers have achieved state-of-the-art performance on various computer vision benchmarks, demonstrating their effectiveness in understanding and modeling visual data.

### 1.2. Evaluating Artificial Intelligence Algorithms

In the field of dermatological AI, evaluation metrics play a crucial role in assessing algorithm performance. The Area Under the Receiver Operating Characteristic curve (AUROC or AUC) stands as the predominant evaluation metric, quantifying an algorithm’s discriminative ability between positive and negative cases. A perfect AUROC score of 1.00 indicates optimal discrimination, while 0.5 signifies discrimination by chance, equivalent to random guessing [14,15]. The ROC curve offers the user ability to assess the algorithm at different sensitivity and specificity operating points, enabling them to manage the decision thresholds for different diagnostic applications. Sensitivity measures the algorithm’s ability to correctly identify true positive cases, while specificity evaluates its accuracy in identifying true negatives. Complementary metrics, including precision and F1 score, also provide additional dimensions of performance assessment. Precision quantifies the accuracy of positive predictions, and the F1 score balances precision and recall. This comprehensive suite of metrics enables a nuanced evaluation of AI algorithms, offering insights into their capacity to accurately classify cases, the reliability of their predictions, and the inherent trade-offs between different performance aspects. Such thorough evaluation is essential for understanding an algorithm’s potential clinical utility and limitations in dermatological applications. For segmentation tasks, the DICE coefficient [16] or the Jacard index [17] are the most widely used evaluation metrics. These metrics quantify the overlap between two sets, ranging from zero (no overlap) to one (perfect overlap). The Jaccard coefficient measures similarity by comparing set intersection to union, frequently used in text analysis and image segmentation. The Dice coefficient, while similar, weights commonalities more heavily than differences.

## 2. Datasets

Datasets have been created to evaluate, validate, and enhance algorithms. Small datasets restrict the learning and generalizability capabilities of algorithms. Thus, the availability of large, demographically expansive, and standardized datasets is essential [18]. Numerous datasets comprising clinical and/or dermoscopic images are available, and their number continues to grow.

### 2.1. ISIC Archive

The ISIC Dataset was developed by the International Skin Imaging Collaboration (ISIC) to advance digital imaging systems and decrease mortality rates from skin cancer [19]. The initial version of the dataset (ISIC’16) which comprises 900 training and 379 test samples was introduced at the International Symposium on Biomedical Imaging (ISBI) 2016 challenge. The samples in the dataset are categorized into two classes, melanoma, and nevus, with roughly 30.3% of the images allocated to the melanoma category while the remaining categorized as nevus.

ISIC continuously expands its image collection and releases machine learning challenge datasets regularly. In the ISIC’17 dataset, in addition to melanoma and nevus, images of seborrheic keratosis are incorporated. This dataset comprises 2000 training images (comprising 374 melanoma, 254 seborrheic keratoses (SK), and 1372 nevi), 150 validation images (with 30 melanoma, 42 SK, and 78 nevi), and 600 testing images (featuring 117 melanoma, 90 SK, and 393 nevi). In the ISIC’18 dataset, the classes are diversified, encompassing 12,594 training images, 100 validation images, and 1000 testing images, spanning eight distinct skin lesion categories, including melanoma, melanocytic nevus, basal cell carcinoma (BCC), actinic keratosis (AK), benign keratosis, dermatofibroma, vascular lesion, and squamous cell carcinoma (SCC). The subsequent ISIC’19 dataset incorporates additional metadata such as age, gender, anatomical region, and the gold standard lesion diagnosis. ISIC 2019 comprises 25,331 training images and 8239 test images. Expanding on this, ISIC’20 [20] integrates metadata with patient ID, similar to the ISIC 2019 dataset. It contains 33,126 training images and 10,982 test images. The images within the ISIC dataset originate from diverse geographical regions, including Spain, Australia, Austria, the United States, Greece, Turkey, New Zealand, Sweden, and Argentina.

Most recently, ISIC released the SLICE-3D (“Skin Lesion Image Crops Extracted from 3D Total Body Photography (TBP)”) dataset that comprises ~400 K standardized cropped images of lesions from 3D Total Body Photography (TBP) [21]. The images in this dataset mimic non-dermoscopic close-up images of lesions covering 15 mm × 15 mm of field of view. This dataset is used as the training data for the most ISIC’24 Kaggle challenge, which was attempted by ~3500 participants worldwide with ~80 K submissions. All the ISIC datasets are publicly available in the ISIC Archive [19].

### 2.2. HAM10000

The HAM10000 dataset [22], short for the “Human Against Machine” dataset, is a publicly accessible dataset comprising images sourced from Cliff Rosendahl’s skin cancer clinic in Queensland, Australia and the Department of Dermatology at the Medical University of Vienna, Austria. This dataset encompasses 10,015 dermoscopic images representing seven types of skin conditions: 327 images of actinic keratosis, 514 images of basal cell carcinoma, 1099 images of benign keratosis, 115 images of dermatofibroma, 1113 images of melanocytic nevi, 6705 images of melanoma, and 142 images of vascular skin lesions. Along with the images, patients’ age and gender information are included in the dataset [22].

### 2.3. PH2

The PH2 dataset [23] comprises dermoscopic images gathered at the Dermatology Center of Pedro Hispano Hospital in Portugal. It comprises 200 dermoscopic images, including 80 nevus, 80 atypical nevus, and 40 melanoma images. This dataset includes medical annotations for the lesion images, covering aspects like medical segmentation of pigmented skin lesions, histological and clinical diagnoses, and evaluation of different dermoscopic criteria like asymmetry, dots/globules, streaks, colors, regression, pigment network, and blue-whitish veil. Notably, patient metadata are absent from this dataset [23].

### 2.4. DERMOFIT Image Library: Edinburgh Dataset

The DERMOFIT Image Library (Edinburgh Dataset) [24,25] comprises 1300 high-quality clinical skin lesion images gathered by the University of Edinburgh under standardized conditions. These lesions are categorized into ten distinct classes: AK (45 images), BCC (239 images), melanocytic nevus (331 images), SK (257 images), SCC (88 images), intraepithelial carcinoma (78 images), pyogenic granuloma (24 images), hemangioma (97 images), dermatofibroma (65 images), and malignant melanoma (76 images). Each image underwent diagnosis utilizing the gold standard method, which involved expert evaluations by dermatologists and dermatopathologists. Additionally, each lesion is accompanied by a binary segmentation mask that denotes the area encompassing the lesion. The dataset predominantly comprises images of patients of Caucasian descent [24,25].

### 2.5. BCN20000

The BCN200000 dataset [26] comprises 19,424 dermoscopic images acquired at the Department of Dermatology of the Hospital Clínic in Barcelona. It encompasses nine classes: nevus, melanoma, basal cell carcinoma, seborrheic keratosis, actinic keratosis, squamous cell carcinoma, dermatofibroma, vascular lesion, and “other” lesions not classified in the mentioned categories. Aimed at exploring the issue of unrestricted classification of dermoscopic skin cancer images, this dataset includes lesions situated in challenging diagnostic sites (like nails and mucosa), as well as large lesions exceeding the dermoscopy device’s field of view and hypopigmented lesions. Image editing and filtering using diverse computer vision techniques were applied to eliminate noise, background artifacts, and other imperfections. Additionally, the dataset incorporates metadata regarding the anatomical site of the lesions, as well as the patients’ age and gender, mirroring real-world clinical scenarios [26].

### 2.6. DermQuest

The publicly available DermQuest initially comprised 137 dermoscopic images, consisting of 76 melanoma images and 61 non-melanoma lesion images. The DermQuest dataset was transferred to Derm101 [27] in 2018.

### 2.7. DermIS

The DermIS dataset [28], an abbreviation for the Dermatology Information System, was developed through a collaboration between the Department of Dermatology at the University of Erlangen and the Department of Clinical Social Medicine at the University of Heidelberg. This dataset comprises 7172 images, incorporating 43 annotated macroscopic images of melanoma lesions and 26 macroscopic images of non-melanoma lesions. The dataset also includes age, gender, and anatomical localization information [28].

### 2.8. Asan Dataset

The Asan dataset [29,30], sourced from the Department of Dermatology at the Asan Medical Center, encompasses 17,125 clinical images with 12,656 confirmed by biopsy, distributed across 12 categories: BCC (1082 images), SCC (1231 images), intraepithelial carcinoma (918 images), AK (651 images), SK (1423 images), malignant melanoma (599 images), nevus (2706 images), lentigo (1193 images), pyogenic granuloma (375 images), hemangioma (2715 images), dermatofibroma (1247 images), and wart (2985 images). Patient data in the dataset include information on age, gender, and race, with over 99% of the dataset representing individuals of Asian descent. Furthermore, pathological observations from biopsied patients are documented alongside the image data [29,30].

### 2.9. MED-NODE

The MED-NODE Dataset [31], assembled by the Department of Dermatology at the University Medical Center Groningen (UMCG), comprises 170 macroscopic images depicting cases of melanoma and nevus (comprising 70 cases of superficial spreading melanomas and 100 nevi). The dataset’s patient cohort predominantly comprises individuals with light skin (Caucasian descent). Any artifacts within the dataset were eliminated through manual software intervention [31].

### 2.10. Fitzpatrick 17k

The dataset [32,33] was developed by integrating two publicly available atlases, DermaAmin [34] and Atlas Dermatologico [35], as implemented by Groh et al. It comprises 16,577 images sourced from these atlases, labeled according to skin type. The dataset encompasses a total of 114 skin conditions, with each condition represented by at least 53 images. The majority of images depict fair-skinned patients. When categorized by skin color, the dataset includes 7755 images of the lightest skin types (1 and 2), 6089 images of the medium skin types (3 and 4), and 2168 images of the darkest skin types (5 and 6). The skin conditions in the dataset are classified into three main categories: 2234 benign lesions, 2263 malignant lesions, and 12,080 non-neoplastic lesions. On a more detailed level, the conditions are further categorized into nine specific groups: 10,886 inflammatory lesions, 1352 malignant epidermal lesions, 1194 genodermatoses, 1067 benign dermal lesions, 931 benign epidermal lesions, 573 melanomas, 236 benign melanocytic lesions, 182 malignant cutaneous lymphomas, and 156 malignant dermal lesions [32,33].

### 2.11. SCIN

The SCIN dataset [36] was designed by the Google Research team and Stanford University to enhance the diversity of publicly available dermatology images for use in health education and research. Images were collected from Internet users through advertisements, resulting in a collection of over 10,000 images of dermatological conditions. The dataset also includes metadata and disease labels such as age, gender, ethnicity/race, Fitzpatrick skin type, and Monk skin tone. After collection, the images were processed and de-identified by removing features such as tattoos, facial features, and landmarks. A group of dermatologists labeled the images according to a weighted differential diagnosis. In the dataset, 7.55% of the images were categorized as Fitzpatrick type 1, 40.21% as type 2, 30.76% as type 3, 13.64% as type 4, 5.27% as type 5, and 0.57% as type 6. When categorized by disease type, 5.15% of the conditions were identified as eruptions (comprising inflammatory, reactive, drug-induced, and other types), 21.81% as cutaneous infections, 10.05% as contact dermatitis, 2.24% as vascular conditions, 0.75% as pigmentary disorders, 0.25% as nail conditions, and 0.04% as hair disorders. Additionally, 3.85% of the lesions were classified as benign, while 1.35% were classified as malignant or premalignant [36].

### 2.12. SkinCAP

SkinCAP [37] was initially developed to assist in diagnosing dermatological patients and in improving vision-based large language models by providing a dataset with comprehensive medical annotations. It comprises 4000 images sourced from the Fitzpatrick 17k skin disease dataset [32,33] and the Diverse Dermatology Images dataset [38]. Board-certified dermatologists annotated these images with detailed medical information, including location, distribution, color, morphology, and other pertinent features. The annotations also include the most likely differential diagnosis [37].

### 2.13. SLICE-3D Dataset

This dataset [21] was created to assist non-specialist physicians in diagnosing skin lesions and to support triage in primary care. It consists of macroscopic images of skin lesions cropped from whole-body photographs, totaling over 400,000 images collected from seven clinics: Memorial Sloan Kettering Cancer Center in the USA, Hospital Clínic de Barcelona in Spain, University of Queensland in Australia, Medical University of Vienna in Austria, University of Athens in Greece, Melanoma Institute Australia, and University Hospital Basel in Switzerland. The images are standardized with a consistent device model and field of view. The dataset includes diagnostic labels, patient gender, age at the time of imaging, anatomical location of the lesion, illumination pattern of the 3D TBP image, data from the lesion imager (such as lesion diameter estimate), and for melanoma-consistent biopsies, the depth of invasion and mitotic index. Lesions that underwent biopsy are classified as strongly labeled, with diagnoses recorded. Lesions not biopsied are classified as weakly labeled and recorded as “benign”. All malignant lesions fall into the strongly labeled category due to histopathological confirmation. Some benign lesions are strongly labeled (histopathologically confirmed and recorded as diagnosed), while others are weakly labeled and recorded as benign. Since the dataset is retrospective, it does not include skin tones. It contains 393 images of malignant lesions (163 BCC, 73 SCC, 157 melanoma), 114 indeterminate lesion images (39 actinic keratosis and 75 melanocytic proliferation), and 400,552 benign lesion images (1 apocrine or eccrine adnexal epithelial proliferation, 2 follicular adnexal epithelial proliferation, 83 epidermal proliferation, 443 melanocytic proliferation, 15 fibro-histiocytic, 3 vascular, 2 cyst, 5 flat melanotic pigmentation without melanocytic nevus, and 399,991 NOS) [21].

### 2.14. Diverse Dermatology Images

This dataset [38], expertly curated with pathologically validated diagnosis labels by Stanford University, is designed to train algorithms with validated features. It comprises 656 clinical images from 570 patients and includes information on skin tones, age, and gender. The dataset is specifically structured to compare patients with dark skin tones (Fitzpatrick skin type 5–6) to those with light skin tones (Fitzpatrick skin type 1–2). It contains 208 images of Fitzpatrick skin type 1–2 (159 benign and 49 malignant), 241 images of Fitzpatrick skin type 3–4 (167 benign and 74 malignant), and 207 images of Fitzpatrick skin type 5–6 (159 benign and 48 malignant). The dataset includes images representing 48 different diagnoses [38].

### 2.15. PAD-UFES-20

The dataset [39] was created by a team at the Federal University of Brazil to provide a publicly accessible collection of clinical images. It contains macroscopic photographs and metadata of skin lesions captured using smartphones. The dataset includes 1641 cervical lesions and 2298 images from 1373 patients, encompassing six different diagnoses (three benign and three malignant). Of the lesions, 58.4% were confirmed by biopsy, and all skin cancer cases fall into this category. The database comprises 730 actinic keratoses, 845 basal cell carcinomas (BCCs), 52 malignant melanomas, 244 melanocytic nevi, 192 squamous cell carcinomas (SCCs), and 235 seborrheic keratoses. Metadata details include the lesion’s diagnosis, anatomical location, horizontal and vertical diameter, subjective symptoms (such as itching, pain, or tenderness), whether the lesion has changed, whether it bleeds, whether it is elevated, and whether a biopsy has been performed. Additionally, it includes the patient’s age, gender, Fitzpatrick skin type, parental origin, history of cancer and skin cancer, exposure to pesticides, smoking and alcohol use, and access to mains water and sewage systems [39].

## 3. Artificial Intelligence in the Diagnosis of Melanoma

### 3.1. Utilization of Clinical Images

Melanoma has historically been screened through clinical examination using established visual assessment methods, most notably the ABCDE criteria. The ABCDE criteria focus on five key characteristics of a mole or lesion: asymmetry (one half of the lesion does not match the other), border irregularity (uneven or poorly defined edges), color variation (multiple colors or shades), diameter (usually larger than 6 mm), and evolving (any changes in size, shape, or color over time). These features help clinicians identify suspicious lesions that may require further investigation [40]. Although state-of-the-art diagnostic tools, including non-invasive imaging devices such as dermoscopes and confocal microscopes, have been developed to improve the accuracy of melanoma detection, visual assessment methods are still commonly used in patient skin self-exams and in primary care settings where non-invasive imaging devices are not available. Utilizing AI to enhance the accuracy of visual assessment methods for evaluating pigmented skin lesions with clinical images may contribute to an earlier diagnosis of melanoma.

Nasr-Esfahani et al. utilized a CNN that consisted of two convolutional layers followed by pooling layers and a fully connected layer with the goal of classifying images as benign or melanoma. Preprocessing techniques were also applied to reduce the illumination artifacts (from non-uniform light and/or reflections of incident light from skin) and noise effects (reducing the effects of normal skin’s texture on classification process). The dataset consisted of 170 clinical images (70 melanoma and 100 benign nevus) from the Department of Dermatology at the University Medical Center Groningen. Due to the small sample size, data augmentation techniques were employed such as cropping, scaling, and rotating to generate 6120 images, where 80% of images were used for training and 20% for testing. Their model achieved an accuracy of 81%, specificity of 80%, sensitivity of 81% (18), NPV of 86%, and a PPV of 86% [41].

Moreover, Yap et al. utilized CNN models (ResNet-50, with and without embedding networks) to extract features from both dermatoscopic and clinical macroscopic images. They applied a late fusion technique (embedding networks) to combine features from both modalities and incorporated metadata such as age, gender, and body location to enhance classification performance. Their dataset included 2917 skin lesion cases from five classes (naevus, melanoma, BCC, squamous cell carcinoma (SCC), and pigmented benign keratoses), with each case containing a dermoscopic image, a macroscopic image, and patient metadata. Using macroscopic images with embedding networks, the AUC for melanoma detection was 0.791. This increased to 0.866 when both macroscopic images and dermoscopy were used; however, the AUC was 0.861 when patient metadata were integrated [42]. Additionally, Riazi Esfahani et al. utilized a CNN to analyze 793 dermatologic images—437 of malignant melanoma and 357 benign nevi obtained from Kaggle. Their model achieved an accuracy of 88.6% for melanoma detection, with a specificity of 81.8% and a sensitivity of 97.1%. However, the study’s limitations were noted as variations in image quality and acquisition methods, which may affect the model’s generalizability [43].

Dorj et al. employed a pre-trained CNN model, AlexNet with 11 layers (5 convolutional layers, 3 max-pooling layers, and 3 fully connected layers) to extract and classify features using an ECOC-SVM classifier. Their dataset consisted of 3753 images (2985 for training and 758 for testing) representing four types of skin cancers: actinic keratoses, BCC, SCC, and melanoma (n = 958, 768 training and 190 testing) obtained from related internet sites. For melanoma classification, the model achieved an average accuracy of 0.942, a specificity of 0.9074, and a sensitivity of 0.9783 [44]. Soenksen et al. assessed multiple deep convolutional neural networks (DCNNs) utilizing a dataset of 33,980 images, encompassing melanoma, SCC, BCC, and various benign lesions. A total of 4063 images of suspicious pigmented lesions (SPLs) were included in this dataset of which 2906 images were melanoma. Images were obtained/generated from open-access dermatology repositories, web scraping outputs, and deidentified clinical images from 133 patients at the Hospital Gregorio Marañón (Madrid, Spain). Data were also divided into six different classes: backgrounds, skin edges, bare skin sections, non-suspicious pigmented lesions of low priority (NSPL), NSPL of medium priority, and SPLs. Blob detection algorithm was initially performed to accelerate analysis. Their baseline DCNN model had three convolutional neural networks and utilized 60% of the data as training, 20% as validation, and 20% as testing. Furthermore, they also trained their DCNN on a 10x non-overlapping augmented dataset with class balancing (naug = 300,000). The VGG16 ImageNet pretrained network was applied as transfer learning to their DCNN as another model. Another transfer learning DCNN model based on the ImageNet’s Xception network was also generated to compare to VGG16’s performance. The VGG16 transfer learning DCNN model demonstrated the highest performance, achieving an AUC of 0.935. The overall AUC across the six included classes (AUCmicro) for this model was 0.97 with a sensitivity of 0.903 and a specificity of 0.899. This model was further applied to analyze wide-field images, using a “saliency-based” approach to detect “ugly duckling” lesions—those that are noticeably abnormal compared to other lesions on the same patient. The model exhibited a 96.3% agreement with the consensus of 10 dermatologists; however, this agreement dropped to 82.96% when examining a reduced number of neighboring lesions [45]. Pomponiu et al. employed a deep neural network (DNN) consisting of a CNN with five convolutional layers and two fully connected layers pre-trained on natural images. Additionally, a KNN classifier was applied to distinguish between benign nevi and melanoma lesions. The dataset consisted of 399 images of pigmented skin lesions (217 benign and 182 melanoma) from online dermatology image libraries (DermIS and DermQuest). Their model achieved an accuracy of 0.83, with a specificity of 0.95 and a sensitivity of 0.92 [46]. Han et al. utilized a DL algorithm (ResNet-152) to classify images of 12 skin diseases (BCC, SCC, intraepithelial carcinoma, actinic keratosis, seborrheic keratosis, melanoma, melanocytic nevus, lentigo, pyogenic granuloma, hemangioma, dermatofibroma, wart). The model was evaluated on multiple datasets, including the Asan and Edinburgh datasets. A total of 19,9398 images from the Asan dataset, MED-NODE dataset, and atlas site images were used for training, while 480 images from the Asan and Edinburgh datasets were used for testing. For melanoma detection in the Asan dataset, the AUC, sensitivity, and specificity were 0.96, 0.91, and 0.904, respectively. For the Edinburgh dataset, these values were 0.88, 0.855, and 0.807, respectively. The model demonstrated strong diagnostic performance, comparable to that of dermatologists, with particularly good results on the Asan dataset. However, the slight performance drop on the Edinburgh dataset highlights the impact of demographic and ethnic differences, as well as variations in image contrast, on the algorithm’s effectiveness [30].

Liu et al. constructed a deep learning system (DLS) with Inception-v4 modules to process images and a shallow module to process metadata such as demographic information and medical history with the goal of identifying 26 of the most common skin cases in adults. Their model was not just used to offer a single diagnosis but to also provide a list of top three differential diagnoses. Primary output was classification from 26 skin conditions and “other”, while secondary output was classification from a full list of 419 skin conditions. Data came from teledermatology cases from a practice serving 17 primary care specialist sites from two states. They performed a temporal split of their data where 80% of cases with metadata (64,837 images) were used for training of the DLS while 20% of their data with metadata (validation set A, 14,833 images) was used for validation. Validation Set A was randomly subsampled to generate Validation Set B (3707 images) to compare the DLS performance to that of dermatologists. DLS performance with Validation Set A for top 1 diagnosis accuracy and sensitivity over 26 skin conditions was 0.71 and 0.58, respectively. These values increased to 0.93 for accuracy and 0.83 for sensitivity for top three diagnoses from the 26 skin conditions. These values were lower for both categories when looking at the full list of 413 skin conditions, but still comparable. When using Validation Set B, DLS demonstrated a top one accuracy of 0.66 compared to 0.63 for that of dermatologists from the 26 skin conditions. Top one sensitivity for DLS with 26 skin conditions was 0.56, which was comparable to that of dermatologist at 0.51. Top three accuracy under the same conditions for DLS was substantially higher at 0.9 compared to 0.75 for that of dermatologists. Top three sensitivity for DLS with 26 skin conditions was 0.64, which was also substantially greater than that of dermatologists at 0.49. Top one and three accuracies on the 419 classification were less than that on the 26 classification for both DLS and dermatologist but were still comparable between the two [47].

Sangers et al. conducted a prospective multicenter study to evaluate skin lesions using an app on iOS and Android devices, comparing the app’s outcomes to histopathological diagnoses or clinical assessments made by dermatologists. They collected images of 785 skin lesions collected from 372 patients from dermatology outpatient clinics of the Erasmus MC Cancer Institute and Albert Schweitzer Hospital in the Netherlands. In total, 418 were classified as suspicious (premalignant or malignant) and 367 as benign. The app utilized CNN (version RD-174) to assess the risk of the photographed lesions, categorizing them as low- or high-risk. Overall app sensitivity and specificity were 0.869 and 0.704, respectively. For melanocytic lesions, the sensitivity and specificity were 0.819 and 0.733, respectively. One limitation of the study was that lesion photos were taken by trained researchers in outpatient settings rather than by patients, which may affect the app’s external validity, as it is intended for general use in non-clinical environments. Additionally, the study employed two high-resolution smartphone models, raising concerns about the app’s performance on devices with lower camera resolution or older hardware. Furthermore, over 80% of participants had Fitzpatrick skin types 1 or 2, which may limit the study’s applicability to individuals with darker skin tones. Lastly, the low number of melanoma cases (n = 12) restricts conclusions about the app’s capability to detect melanomas specifically. Despite these important limitations, the study introduces the concept of utilizing smartphone apps for self-skin examination and self-assessment of skin cancer risk, which could be highly beneficial for early detection of skin cancers [48].

Polturu et al. employed an automated machine learning model (AutoML) created using a no-code online service platform to analyze a dataset of 87 non-melanoma images and 119 melanoma images, all taken with a consumer-grade camera and obtained from the DermIS and DermQuest public datasets. The model attained an overall accuracy of 0.844, with a specificity of 0.857 and a sensitivity of 0.833 [49]. Algorithms used in the diagnosis of melanoma from clinical images are summarized in Table 1.

### 3.2. Utilization of Dermoscopic Images

Dermoscopy is currently used as a non-invasive diagnostic measure of skin lesions. It is particularly useful in the differential diagnosis of skin tumors [50]. Recently, AI models and technologies have been applied to dermoscopic imaging, and successful results have been obtained in the differential diagnosis of skin tumors. We reviewed 37 studies that used dermoscopic images as a dataset (Table 2 and Table 3). Of these 37 studies, 26 evaluated AI models in the diagnosis of melanoma and 11 studies evaluated both melanoma and non-melanoma skin cancers.

#### 3.2.1. Distinguishing Melanoma from Benign Lesions

Masood et al. classified clinical and dermoscopic photographs as benign/melanoma using ANN and compared the performances of three different ANN algorithms (Levenberg-Marquardt (L-M), resilient backpropagation (RP), scaled conjugate gradient (SCG)). SCG offered the most successful results with 92.6% sensitivity and 91.4% specificity. LM achieved a specificity of 95.1% in benign lesions, but it was not as successful as SCG in melanoma [51]. In [52], a fusion ML model consisting of five individual top-ranked algorithms from the ISBI 2016 Challenge was applied for melanoma detection, and its performance was compared to dermatologists. Their model achieved an AUROC of 0.86 and was more accurate than dermatologists; however, applying ML classifications to dermatologist evaluations increased dermatologist sensitivity from 76.0% to 80.8% and specificity from 72.6% to 72.8% [53].

Recently, there has been a surge in studies on the discrimination of melanocytic lesions (benign/malignant) using CNNs on dermoscopic photographs. Chanki Yu et al. used a pre-trained CNN model (VCG-16) in diagnosing acral melanoma compared to both general practitioners and dermatologists. They performed two-fold cross-validation and split the dataset into a 50/50 train–test split. The model achieved a similar AUROC value to experts and was significantly superior to the non-expert group [54]. Abbas et al. also designed a seven-layer deep CNN to discriminate between acral melanoma and benign nevus. They used 724 dermoscopic images from Chanki Yu et al.’s [54] dataset and 4344 dermoscopic images generated by data augmentation techniques. The authors also applied transfer learning to the AlexNet and ResNet-18 and fine-tuned them by modifying their last layers. An AUC of 0.97, 0.96, and 0.91 was obtained with ResNet-18, AlexNet, and the proposed ConvNet, respectively [55]. Another study proposed a CNN model to distinguish combined nevi from melanoma. Moleanalyzer Pro, previously trained on more than 120,000 dermoscopic images, was used in the study, and 72 dermoscopic images (36 combined nevus and 36 melanoma) were evaluated. When compared to 11 dermatologists divided into three groups (beginner/qualified/expert), the model outperformed all of them, revealing 97.1% sensitivity and 78.8% specificity [56].

Even though AI has shown small initial success against the participating dermatologist in [56], only a limited number of dermatologist were included. To address this drawback, Brinker et al. compared the performance of a CNN algorithm (Resnet) trained only on open-source dermoscopic images with 157 dermatologists, resulting in seven dermatologists being more accurate than CNNs [57]. Furthermore, Giulini et al. combined CNN and human expertise in the diagnosis of melanoma. In the study, 64 physicians (33 dermatologists, 11 dermatology residents, and 20 general practitioners) assessed 100 dermoscopic photographs of 50 melanomas and 50 benign nevi. After a duration of 4 months, the same photographs were reevaluated in a different order with CNN assistance by the physicians. In the session with CNN assistance, the mean sensitivity and specificity increased to 67.88% and 73.72% from 56.31% and 69.28%, respectively [58].

Hybrid models are also commonly studied in the literature; Mahbod et al. used three pre-trained CNN models (AlexNet, VGG16, and ResNet-18) for feature extraction, followed by an SVM-based classification step. The final classification result was obtained by averaging the output of the individual models. The resulting ensemble model was evaluated on 150 validation images and achieved an AUC of 90.69%, surpassing the performance of the individual CNN models (AlexNet, VGG16, and ResNet-18) [59]. Ningrum et al. constructed a hybrid model by integrating dermoscopic pictures and patient data to diagnose melanoma. They employed a model that utilized both CNNs to analyze photos and ANNs to analyze patient data to categorize patients as melanoma or nonmelanoma. The results were compared with CNNs analyzing images only. The CNN+ANN model achieved an accuracy of 92.34%, surpassing the accuracy of the CNN model alone at 73.69% [60].

While AI demonstrates success in studies, its application and implementation in real-world scenarios are crucial. Hekler et al. assessed the efficacy of DL in categorizing lesions by employing multiple real-world lesion images, single lesion images, and modified lesion images. The model displayed markedly enhanced performance when utilizing multiple real-world images, particularly in uncertainty estimation and robustness [61]. Specifically, the utilization of AI in melanoma screening is poised to substantially alleviate the workload on clinicians. To showcase AI’s potential as a melanoma screening tool, Crawford et al. explored the feasibility of employing AI to identify potential melanomas in self-referred patients concerned about the malignancy of their skin lesions. The AI successfully identified 11 of 17 malignant lesions, achieving an accuracy of 73.56%, exceeding the accuracy of four out of five dermatologists involved in the study [62].

The lack of transparency of AI techniques reduces their reliability for users. To address this issue, Chanda et al. developed an explainable AI (XAI) algorithm. In the task of predicting melanoma, the algorithm explains the basis of its prediction. The investigation revealed that the XAI increased clinicians’ diagnostic confidence while also enhancing their trust in the assistance provided by XAI [63]. Correira et al. introduced a method that utilizes an interpretable prototypical-part model that integrates binary masks, automatically generated by a segmentation network and user-refined prototypes. This model is designed to incorporate non-expert feedback, ensuring that the learned prototypes specifically relate to important areas within the skin lesion while excluding irrelevant factors beyond its boundaries. By following these two distinct information pathways, the proposed approach demonstrates superior diagnostic performance when compared to non-interpretable models [64].

**Table 2 life-14-01602-t002:** Algorithms that distinguish between melanoma and benign lesions through dermoscopic images.

Publication	End-Point	Dataset	Algorithm	Performance
Masood et al. [51]	Classification (benign/melanoma)	135 images (Clinical + dermoscopic) 107 for training, 14 for validation 14 for testing. Images were obtained from one clinic in France; the ethnicity and skin types were not specified.	Compared 3 ANN algorithms (RP, L-M, SCG)	**SCG:** Acc: 91.9% Sen: 92.6% Spe: 91.4% **L-M:** Acc: 91.1% Sen: 85.2% Spe: 95.1% **RP:** Acc: 88.1% Sen: 77.8% Spe: 95.1%
Aswin et al. [65]	Classification (Cancerous/Non-cancerous)	30 dermoscopic images for training. 50 dermoscopic images for testing. No further information regarding the dataset was provided.	Hybrid Genetic Algorithm + ANN	Acc: 88%
Xie et al. [66]	Classification (MM/BN)	Dermoscopic images Xanthous race: 240 images (80 MM, 160 BN). Caucasian race: 360 images (120 MM, 240 BN). Images were obtained from a clinic in China.	Proposed: meta-ensemble model of multiple neural network ensembles Ensemble 1: single-hidden-layer BP nets with the same structures Ensemble 2: single-hidden-layer BP nets and fuzzy nets Ensemble 3: double-hidden-layer BP nets with different structures	**Xanthous race:** Sen: 95% Spe: 93.75% Acc: 94.17% **Caucasian race:** Sen: 83.33% Spe: 95% Acc: 91.11%
Marchetti et al. [52]	Classification (MM/BN)	ISBI 2016 challenge dataset [67]: MM: 248 images, BN: 1031 images, Train set: 900 images, Test set: 379 images, Reader study: 100 images (50 MM, 50 BN).	Five methods (unlearned and machine learning) were used to combine individual automated predictions into “fusion” algorithms	**Top Fusion Algorithm: Greedy Fusion:** Sen: 58% Spe: 92% AUC: 86% **Dermatologists:** Sen: 82% Spe: 59% AUC: 71%
Marchetti et al. [53]	Classification (MM/BN/SK) and (biopsy/observation)	ISIC archive [19]: 2750 dermoscopy images (521 (19%) MM, 1843 (67%) BN, and 386 (14%) SK). Training set: 2000 images, Validation: 150 images, Test set: 600 images.	ISBI 2017 Challenge top-ranked algorithm	**Algorithm:** Sen: 76% Spe: 85% AUC: 0.87 **Dermatologists:** Sen: 76.0% Spe: 72.6% AUC: 0.74
Cueva et al. [68]	Classification (Cancerous/Non-cancerous)	PH^2^ database [23]: Training set: 30 images (10 MM, 10 common mole, 10 no-common mole). Test set: 201 images (80 common mole, 80 no-common mole, 41 MM).	ANN with backpropagation algorithm	After an analysis of 201 images in the algorithm developed a performance of 97.51% was obtained
Navarro et al. [69]	Segmentation and registration to evaluate lesion change	ISIC archive [19]: Training set: 2000 dermoscopic images. Validation: 150 dermoscopic images. Test set: 600 dermoscopic images.	Segmentation: LF-SLIC Registration: SP-SIFT	Acc: 0.96 for segmentation
Yu C. et al. [54]	Classification (melanoma/non-melanoma)	725 images obtained from two clinics in South Korea. The ethnicity and skin types were not specified. (AM: 350 images, BN: 374 images). Group A: 175 images. AM, 187 images BN. Group B: 175 images. AM, 187 images BN. Training set: Group A images for training Group B. Group B images for training Group A. Test set: Group A images for Group A. Group B images for Group B.	CNN (VCG-16)	**Group A:** **CNN:** Sen: 92.57% Spe: 75.39% Acc: 83.51% **Expert:** Sen: 94.88% Spe: 68.72% Acc: 81.08% **Non-expert**: Sen: 41.71% Spe: 91.28% Acc: 67.84% **Group B:** **CNN:** Sen: 92.57% Spe: 68.16% Acc: 80.23% **Expert:** Sen: 98.29% Spe: 65.36% Acc: 81.64% **Non-expert**: Sen: 48.00% Spe: 77.10% Acc: 62.71%
Abbas et al. [55]	Classification (benign nevus/acral melanoma)	724 images from Yonsei University, South Korea. The ethnicity and skin types were not specified [54] (350 acral melanoma, 374 benign nevi). 4344 images with data augmentation (2100 acral melanoma, 2244 benign nevi).	Compared three CNN algorithms (Seven-layered deep CNN, ResNet-18, AlexNet)	**ResNet-18** Acc: 0.97 AUC: 0.97 **AlexNet:** Acc: 0.96 AUC: 0.96 **Proposed ConvNet** Acc: 0.91 AUC: 0.91
Fink et al. [56]	Classification (Benign/Malignant)	Training set: >120,000, dermoscopic images and labels. Test set: 72 images (36 combined naevi, 36 melanomas). Images were obtained from three clinics in Germany; the skin types and ethnicity were not specified.	CNN (Moleanalyzer-Pro) based on a GoogleNet Inception_v4 architecture	**CNN:** Sen: 97.1% Spe: 78.8% **Dermatologists:** Sen: 90.6% Spe: 71.0%
Phillips et al. [70]	Classification (MM/dysplastic nevi/other)	Pretrained algorithm Training set (in study): 289 images (36 melanoma lesions; 67 nonmelanoma lesions, 186 control lesions). Test set: 1550 images Images were obtained from three clinics in Germany; the ethnicity and skin types were not specified.	SkinAnalytics (CNN)	**The algorithm:** **İphone 6s image**: AUC: 95.8% Spe: 78.1% **Galaxy S6 image:** AUC: 93.8% Spe: 75.6% **DSLR image:** AUC: 91.8% Spe: 45.5% **Specialists:** AUC: 77.8% Spe: 69.9%
Martin-Gonzalez et al. [71]	Classification (benign/ malignant skin lesion)	Pretrained with 37,688 images from ISIC archive [19] 2019 and 2020. Training set: 339 images (143 MM, 196 BN). Test set: 232 images (55 MM, 177 BN). Test set images were obtained from the clinic in Spain. The images used in the study were of light-skinned patients.	QuantusSKIN (CNN)	AUC: 0.813 Sen: 0.691 Spe: 0.802 Acc: 0.776
Brinker et al. [57]	Classification (Melanoma/Nevi)	Training set: 12,378 dermoscopic images from the ISIC dataset [19]. Test set: 100 dermoscopic images (20 MM, 80 Nevi).	ResNet-50 (CNN)	**Algorithm:** Sen: 74.1% Spe: 86.5% **Dermatologists:** Sen: 74.1% Spe: 60%
Giulini et al. [58]	Classification (Melanoma/Nevi)	Over 28,000 dermoscopic images; the ethnicity and skin types of the training set were not specified. CNN test set: 2489 images (344 melanomas, 2155 nevi). Physician test set: 100 images (50 MM, 50 nevi). The test set consisted of images of patients with Fitzpatrick skin types 1–4.	Session 1: Physicians without CNN Session 2: Physicians with CNN	**Physicians without CNN** Sen: 56.31% Spe: 69.28% **Physicians with CNN** Sen: 67.88% Spe: 73.72%
Ding et al. [72]	Classification (Binary: melanoma/non-melanoma and multiclass: benign nevi, seborrheic keratosis or melanoma)	ISIC dataset [19]: Training set: 2000 images (374 MM, 254 SK, 1372 BN). Validation set: 150 images (30 MM, 42 SK, 78 BN). Test set: 600 images (117 MM, 90 SK, 393 BN).	Segmentation: U-Net Classification: Five CNNs (Inception-v3, ResNet-50, Densenet169, Inception-ResNet-v2, and Xception) with SE-block and the neural network for ensemble learning consisting of two local connected layers and a softmax layer	**Binary:** **Inception-v3** Acc: 0.885 AUC: 0.883 **ResNet-50** Acc: 0.88 AUC: 0.882 **Densenet169** Acc: 0.893 AUC: 0.882 **Inception-ResNet-v2** Acc: 0.89 AUC: 0.894 **Xception** Acc: 0.891 AUC: 0.896 **Ensemble** Acc:0.909 AUC: 0.911 **Multiclass:** **Inception-v3** Acc: 0.792 AUC: 0.883 **ResNet-50** Acc: 0.762 AUC: 0.864 **Densenet169** Acc: 0.800 AUC: 0.881 **Inception-ResNet-v2** Acc: 0.800 AUC: 0.873 **Xception** Acc: 0.810 AUC: 0.896 **Ensemble** Acc: 0.851 AUC: 0.913
Yu L. et al. [73]	Segmentation and Classification (Benign/Malignant)	ISIC dataset [19]: Training set: 900 images. Test set: 350 images.	FCRN for skin lesion segmentation and very deep residual network for classification	**Segmentation:** Sen: 0.911 Spe: 0.957 Acc: 0.949 **Classification with segmentation:** Sen: 0.547 Spe: 0.931 Acc: 0.855
Bisla et al. [74]	Classification (Nevus, SK, MM)	Training set: ISIC dataset [19]: 803 MM, 2107 nevus, 288 SK. PH^2^ dataset [23]: 40 MM, 80 Nevus Edinburgh dataset. [25]: 76 MM, 331 nevus, 257 SK. Test set: ISIC data sets 600 images (117 MM, 90 SK, and 393 nevus),	Segmentation: Modified U-Net (CNN) Augmentation: de-coupled DCGANs Classification:ResNet-50	AUC: 0.915 Acc: 81.6%
Mahbod et al. [59]	Classification (MM/All, SK/All)	ISIC dataset [19]: Training: 2037 dermoscopic images (411 MM, 254 SK, 1372 BN).	Feature Extraction: Pretrained CNNs (AlexNet, ResNet-18 and VGG16) Classification: SVM	AUC: 90.69
Bassel et al. [75]	Classification (Benign/Malignant)	ISIC dataset [19]: 1800 images of benign type and 1497 pictures of malignant cancer. Training set: 70% of images (1440 benign, 1197 malignant). Test set: 30% of images (360 benign, 300 malignant).	Model 1:Feature Extraction: ResNet50 Model 2:Feature Extraction: VCG-16 Model 3:Feature Extraction: Xception Classification: Stacked CV model (SVM+NN+RF+KNN)	**ResNet Model:** Acc: 81.6% AUC: 0.818 **VCG-16 Model:** Acc: 86.5% AUC: 0.843 **Xception Model:** Acc: 90.9% AUC: 0.917
Ningrum et al. [60]	Classification (Melanoma/benign)	ISIC dataset [19]: 900 images. Training set: 720 images. Validation set: 180 images. Test set: 300 (93 malignant, 207 nonmalignant).	Classification: CNN model for images + ANN model for patient metadata	**CNN** Acc: 73.69 AUC: 82.4 **CNN+ANN** Acc: 92.34 AUC: 97.1
Nambisan et al. [76]	Segmentation and classification (Melanoma/Benign)	ISIC dataset [19]: Segmentation task: 487 MM images. Classification task: 1000 images (500 MM, and 500 benign (100 images per class from the Actinic keratosis, Melanocytic nevus, Benign keratosis, Dermatofibroma, and Vascular lesion).	Segmentation (Classification dataset+Segmentation dataset (Irregular networks)) U-Net/U-Net++/MA-Net/PA-Net Handcrafted Feature Extraction Classification: Level 0 (without segmentation): DL classification model Level 1 (With segmentation and with level 0 model’s results): Conventional classification model	**Conventional Ensemble** Acc: 0.793 **DL Ensemble** Acc: 0.838 **EfficientNet-B0 + Conventional** **Ensemble** Acc: 0.862
Collenne et al. [77]	Classification (Melanoma/Nevi)	ISIC dataset [19]: (6371 nevi and 1301 melanoma) Training set: 70% of images. Validation set: 10% of images. Test set: 20% of images.	Segmentation: U-Net Classification ANN (for asymmetry features + CNN (EfficientNet)	**Handcrafted Model with asymmetry features (ANN):** Acc: 79% AUC: 0.87 Sen: 90% Spe: 67% **ANN+CNN:** Sen: 0.92 Spe: 0.82 Acc: 0.87 AUC: 0.942
Hekler et al. [61]	Classification (Melanoma/Nevi)	HAM10000 [22] and BCN20000 [26] datasets: 29,562 images (7794 melanoma and 21,768 nevi). 80% training, 20% validation Test set: SCP2 dataset, 293 melanoma and 363 melanocytic nevi from 617 patients.	ConvNeXT architecture 1. Classification using a single image 2. Classification using multiple real-world images 3. Classification using multiple artificially modified images	**Single image approach:** Acc: 0.905 ECE: 0.131 **Multiview real-world approach:** Acc: 0.930 ECE: 0.072 **Multiview artificial approach:** Acc: 0.929 ECE: 0.086
Crawford et al. [62]	Classification (Excision/no excision)	Self-referred patients: The test set consisted of patient images, the majority of whom were of Scottish and Irish descent, mostly Fitzpatrick skin types 1, 2, and 3.	MoleAnalyzer Pro	**AI** Sen: 64.7% Spe: 75.76% PPV: 40.0% NPV: 89.6% Acc: 73.56%

Artificial Neural Network (ANN); Levenberg–Marquardt (L-M); Resilient Back-propagation (RP); Scaled Conjugate Gradient (SCG); Accuracy (Acc); Sensitivity (Sen); Specificity (Spe); Malign Melanoma (MM); Benign Nevi (BN); Back-propagation (BP); International Symposium on Biomedical Imaging (ISBI) challenge 2016; Area under the ROC curve (AUC); International Skin Imaging Collaboration (ISIC); Seborrheic Keratosis (SK); Local Features—Simple Linear Iterative Clustering (LF-SLIC); Scale Invariant Feature Transform (SIFT); Acral Melanoma (AM); Convolutional Neural Network (CNN); Visual Geometry Group (VGG), Squeeze-and-Excitation block (SE-Block); Fully Convolutional Residual Network (FCRN); Deep Convolutional Generative Adversarial Network (DCGAN); Neural network (NN); Random forest (RF); Human Against Machine with 10000 training images (HAM10000); Expected calibration error (ECE); Artificial Intelligence (AI); Negative predictive value (NPV); Positive predictive value (PPV).

#### 3.2.2. Distinguishing Melanoma from Other Skin Cancers

Esteva et al. used a pre-trained GoogLeNet Inception v3 architecture and performed transfer learning on 127,463 clinical images, including 3374 dermoscopy images containing 2032 diseases. After the CNN model was trained, comparisons were made on 135 epidermal (65 malignant, 70 benign), 130 melanocytic (33 malignant, 97 benign), and 111 melanocytic-dermoscopic (71 malignant, 40 benign) images by 21 board-certified dermatologists. CNN performed on par with dermatologists on all three criteria. The AUC from clinical photographs was 0.94 for melanoma and 0.91 for melanoma from dermoscopic photographs [78]. Rezvantalab et al. also used pre-trained models (DenseNet 201, ResNet 152, Inception v3, InceptionResNet v2) in the classification of eight diagnostic categories (melanoma, melanocytic nevus, BCC, benign keratosis, actinic keratosis, intraepithelial carcinoma, dermatofibroma, vascular lesions, and atypical nevus). All of the models performed better than dermatologists in detecting melanoma and BCC. The most successful model was ResNet 152 with 94.4% AUC in melanoma [79]. Maron et al. included more dermatologists in their study and found that CNNs outperformed dermatologists on both endpoints except BCC [80]. In another study, Tschandl et al. evaluated the success of CNNs in nonpigmented cancers, the most common skin cancer manifestation. They trained the model with dermoscopic and clinical images and compared it to 95 human raters. The evaluators were divided into three groups: beginner, intermediate, and expert, according to their dermoscopy experience. The model’s AUC was higher than the human rating; however, it was less accurate than experts [81]. Tschandl et al. then evaluated the success of ML in benign and malignant pigmented skin lesions. They compared the top three algorithms of the ISIC 2018 challenge with human readers and experts and ultimately outperformed both groups [82]. However, these studies only include images of the lesions and do not include clinical information, which has a very important impact on diagnosis. Therefore, a two-level comparison study including textual information was conducted by Haenssle et al. At Level I, only dermoscopic images were used, while at Level II, clinical and dermoscopic images and textual information were used. At Level I, CNN achieved a higher accuracy than dermatologists, but at Level II, dermatologists achieved a higher accuracy rate [83].

Although dermatologists and AI are seen as competitors in studies, superior results are often recorded with the combination of classifiers. Hekler et al. investigated the potential benefit of combining human and AI data in skin cancer classification. The primary endpoint was the correct classification of images into five designated categories, while the secondary endpoint was the classification of lesions as benign or malignant. Ultimately, the combination of humans and machines achieved 82.95% accuracy; this was 1.36% higher than the best of the two individual classifiers [84]. In Felmingham et al. [85], the authors compared clinicians’ pre- and post-intervention performance when a CNN model was used as an aid. Their results show that residents who are the most inexperienced in the reader group benefited the most, while the experienced dermatologists benefited the least. They also observed that even if use of AI has positive impact in terms of sensitivity of residents. It also led to some unnecessary biopsies. Barata et al. developed a reinforcement learning (RL)-based multiclass classification model incorporating a reward table created by dermatologists to prioritize skin cancer types [86]. This model achieved significantly higher sensitivity for melanoma (79.5%) and basal cell carcinoma (87.1%) compared to a baseline supervised learning model while maintaining 79.2% accuracy. The increased sensitivity was attributed to the RL model’s targeted adjustments influenced by the reward function. In a reader study with 89 dermatologists, those using the RL model as an aid had a mean correct diagnosis score improvement of +12.0%, demonstrating the value of AI support in clinical setting.

**Table 3 life-14-01602-t003:** Algorithms that distinguish melanoma from other skin cancers through dermoscopic images.

Publication	End-Point	Dataset	Algorithm	Performance
Esteva et al. [78]	Classification Binary: Keratinocyte carcinoma/SK; melanoma/nevi 3-way: Benign/Malign/Non-neoplastic 9-way: Cutaneous lymphoma and lymphoid infiltrates/Benign dermal tumors, cysts, sinuses/Malignant dermal tumor/Benign epidermal tumors, hamartomas, milia, and growths/Malignant and premalignant epidermal tumors/Genodermatoses and supernumerary growths/Inflammatory conditions/Benign melanocytic lesions/Malignant Melanoma	ISIC [19] and Edinburgh dataset [25] and the Stanford Hospital: 129,450 clinical images, including 3374 dermoscopic images of 757 disease classes Training set: 127,463 images Test set: 1942 images	Google Inception v3 (CNN)	**Binary classification (Algorithm AUC)** Carcinoma AUC: 0.96 Melanoma AUC: 0.94 Melanoma (Dermoscopic images) AUC: 0.91 **3-way classification:** Dermatologist 1 Acc: 65.6% Dermatologist 2 Acc: 66.0% CNN Acc: 69.4 ± 0.8% CNN partitioning algorithm Acc: 72.1 ± 0.9% **9-way classification:** Dermatologist 1 Acc: 53.3% Dermatologist 2 Acc: 55.0% CNN Acc: 48.9 ± 1.9% CNN partitioning algorithm Acc: 55.4 ± 1.7%
Rezvantalab et al. [79]	Classification (MM/Melanocytic Nevi/BCC/AKIEC/Benign keratosis/DF/Vascular lesion)	HAM10000 dataset [22]: 10,015 dermoscopic images (1113 MM, 6705 nevi, 514 BCC, 327 AK and intraepithelial carcinoma (AKIEC), 1099 benign keratosis, 115 DF, 142 vascular lesions) PH^2^ set [23]: 80 nevi, 40 MM Training set: 70% Validation set: 15% Test set: 15%	Compared CNNs for classification: Inception v3/InceptionResNet v2/ResNet 152/DenseNet 201	**AUC (Melanoma)** Dermatologist AUC: 82.26 DenseNet 201 AUC: 93.80 ResNet 152 AUC: 94.40 Inception v3 AUC: 93.40 InceptionResNet v2 AUC: 93.20 **AUC (BCC)** Dermatologist AUC: 88.82 DenseNet 201 AUC: 99.30 ResNet 152 AUC: 99.10 Inception v3 AUC: 98.60 InceptionResNet v2 AUC: 98.60
Maron et al. [80]	Classification 2-way: Benign/Malignant 5-way: AKIEC/BCC/MM/Nevi/BKL (benign keratosis, including seborrhoeic keratosis, solar lentigo and lichen planus like keratosis)	Training set: 11,444 images (ISIC Archive [19] and HAM10000 dataset [22]) Test set: 300 test images (60 for each of the five disease classes) (HAM10000 dataset)	CNN (ResNet50)	**Two-way classification:** CNN AUC: 0.928 CNN Spe: 91.3% Dermatologist Spe: 59.8% **Five-way classification:** CNN AUC: 0.960 CNN Spe: 89.2% Dermatologist Spe: 98.8%
Tschandl et al. [81]	Classification (Benign/Malignant)	Training set: 7895 dermoscopic and 5829 close-up images Test set: 2072 dermoscopic and close-up images	Combined convolutional neural network (cCNN) (InceptionResNetV2, InceptionV3, Xception, ResNet50)	**cCNN:** AUC: 0.695 Sen: 80.5% Spe: 53.5% **Human Raters:** AUC: 0.742 Sen: 77.6% Spe: 51.3%
Tschandl et al. [82]	Classification (7-way classification: intraepithelial carcinoma including AK and Bowen’s disease; BCC; benign keratinocytic lesions including solar lentigo, SK, and LPLK; dermatofibroma; melanoma; melanocytic nevi; and vascular lesions)	HAM10000 Dataset [22] Training set: 10,015 dermoscopic images Test set: 1195 images	Top 3 algorithms of the ISIC 2018 challenge [87]	**Algorithms (mean):** Sen: 81.9% Spe: 96.2% **Human readers (mean):** Sen: 67.8% Spe: 94.0%
Haenssle et al. [83]	Classification (Benign/Malignant) Management decision (treatment/ excision, no action, follow-up examination)	Pretrained CNN Test set: 100 images including pigmented/ non-pigmented and melanocytic/non-melanocytic skin lesions Dermatoscopic images were collected from several collaborating dermatologists and the ISIC archive [19]. The ethnicity and skin type of patients from whom images were obtained were not specified	Inception v4/Moleanalyzer Pro (CNN)	**CNN Management Decision:** Sen: 95.0% Spe: 76.7% Acc: 84.0% AUC: 0.918 **CNN Diagnosis (Benign/Malignant)** Sen: 95.0% Spe: 76.7% Acc: 84.0% **Level 1 Management Decision:** Dermatologist: Sen: 89.0% Spe: 80.7% Acc: 84.0% **Level 1 Diagnosis (Benign/Malignant)** Dermatologist: Sen: 83.8% Spe: 77.6% Acc: 80.1% **Level 2 Management Decision:** Dermatologist: Sen: 94.1% Spe: 80.4% Acc: 85.9% **Level 2 Diagnosis (Benign/Malignant)** Dermatologist: Sen: 90.6% Spe: 82.4% Acc: 85.7%
Hekler et al. [84]	Primary endpoint: Classification to 5 categories (MM/nevus/BCC/AK, Bowen’s disease or squamous cell carcinoma/seborrhoeic keratosis, lentigo solaris or lichen ruber planus) Secondary end-point: Binary classification (Benign/malignant)	HAM10000 Dataset [22] and ISIC dataset [19] Training set: 12,336 dermoscopic images (585 images of AK, Bowen, SCC, 910 images of BCC, 3101 images of seborrhoeic keratosis, lentigo solaris, lichen ruber planus, 4219 images of nevi, 3521 images of MM)	CNN (ResNet50)	**Multiclass classification:** Physician Acc: 42.94% CNN Acc: 81.59% Physician+CNN Acc: 82.95% **Binary classification:** Physician: Sen: 66% Spe: 62% CNN: Sen: 86.1% Spe: 89.2% Physician+CNN: Sen: 89% Spe: 84%
Xinrong Lu et al. [88]	Classification (normal, carcinoma, and melanoma)	HAM10000 dataset [22] Training set: 8012 images (%80) Test set: 2003 images (%20)	Proposed Xception (The ReLU activation function of the model was replaced with the swish activation function) compared with VGG16, InceptionV3, AlexNet and Xception	VGG16: Acc: 48.99 Sen: 53.7 InceptionV3: Acc: 52.99 Sen: 53.99 AlexNet: Acc: 75.99 Sen: 76.99 Xception: Acc: 92.90 Sen: 91.99 Proposed Xception: Acc: 100 Sen: 94.05
Mengistu et al. [89]	Classification (BCC, SCC, MM)	DermQuest [27] and Dermnet [90] datasets 235 images (162 images for training and 73 images for testing)	Combined SOM and RBFNN and compared them with KNN, ANN, and naïve-Bayes	Proposed modelAcc: 93.15% KNNAcc: 71.23% ANNAcc: 63.01% Naïve-BayesAcc: 56.16%
Rashid et al. [91]	Classification (MM/Melanocytic Nevus/BCC/AKIEC/Benign Keratosis/DF/Vascular Lesion)	ISIC dataset [19] Training set: 8000 images Test set: 2000 images	GAN compared with CNN (DenseNet and ResNet-50)	GAN Acc: 0,861 DenseNet Acc: 0.815 ResNet-50 Acc: 0.792
Alwakid et al. [92]	Classification (MM/BN/BCC/Vascular lesion/Benign keratosis/Actinic Carcinoma/DF)	HAM10000 dataset [22] 10,015 dermoscopic images Training set: 8029 images Validation set: 993 images Test set: 993 images	Inception-V3, InceptionResnet-V2	Inception-V3: Acc: 0.897 Spe: 0.89 Sen: 0.90 InceptionResnet-V2: Acc: 0.913 Spe: 0.90 Sen: 0.91
Felming-ham et al. [85]	Classification (Benign/Uncertain/Malignant)	Training set: 432,390 images from imaging and teledermatology reporting service (ethnicity and skin types were not specified) Version 1 CNN training set: 77.3% Benign and 22.7% malignant Version 2 CNN training set: 78.0% Benign and 22.0% malignant	Version 1: Plain Convolutional Model for pre-intervention period Version 2: Hierarchical deep learning architecture for postintervention period	CNN-sen: 95.8% CNN-spe: 71.5% Teledermatologist sen: 89.5% Teledermatologist-spe: 71.9% CNN-AUC: 0.837 Teledermatologist-AUC: 0.807 Initial resident management plan-AUC: 0.847 AI-assisted resident management plan-AUC: 0.879 Initial teledermatologist management plan-AUC: 0.821
Barata et al. [86]	Classification (MM/BCC/AKIEC/BN/Benign keratinocytic lesions/DF/Vascular lesions)	Training set: HAM10000 dataset 10,015 dermoscopic images Test set: 1511 dermoscopic images; obtained from Austria, Australia, Turkey, New Zealand, Sweden, and Argentina The ethnicity and skin types were not specified	SL Model: ResNet34 Model RL Model: a deep-Q learning model based on a CNN and decides according to the reward system determined by medical experts	**Supervised model:** Sen (Melanoma): 61.4% Sen (BCC): 79.6% Acc: 77.8% **Reinforcement Learning Model:** Sen (Melanoma): 79.5% Sen (BCC): 87.1% Acc: 79.2%

Seborrheic Keratosis (SK); International Skin Imaging Collaboration (ISIC); Convolutional Neural Network (CNN); Area under the ROC curve (AUC); Accuracy (Acc); Malign melanoma (MM); Basal Cell Carcinoma (BCC); Squamous Cell Carcinoma (SCC); Actinic keratosis and intraepithelial carcinoma (AKIEC); Dermatofibroma (DF); Actinic keratosis (AK); Human Against Machine with 10,000 training images (HAM10000); Benign keratosis, including seborrheic keratosis, solar lentigo and lichen planus-like keratosis (BKL); Sensitivity (Sen); Specificity (Spe); Combined convolutional neural network (cCNN); Lichen planus-like keratosis (LPLK); Self-organizing map (SOM); Radial basis function (RBF); Neural network (NN); K-Nearest Neighbors (KNN); Artificial Neural Network (ANN); Generative Adversarial Network (GAN); Supervised Learning (SL); Reinforcement Learning (RL).

### 3.3. In Vivo Skin Imaging Devices

#### 3.3.1. RCM

In recent decades, non-invasive optical imaging methods have been developed to enhance specificity and enable earlier detection of skin cancers. Confocal microscopy (CM) is among these innovative techniques. There are two primary types of CM: reflectance confocal microscopy (RCM) and ex vivo confocal microscopy (EVCM). RCM in particular allows for the in vivo imaging of skin lesions with a “quasi-histologic” resolution, eliminating the need for a biopsy. This imaging technique depends solely on the inherent reflectance contrast of various skin tissue components, without the need for external contrast agents or dyes. As a result, RCM images are presented in grayscale and captured in an en face orientation, in contrast to the “vertical” (i.e., perpendicular to the skin surface) sections commonly used in pathology. RCM has been shown to enhance the specificity and sensitivity in diagnosing melanoma, reduce the number of unnecessary biopsies, and assist in margin assessment and surveillance of melanoma. However, RCM also has certain limitations, such as the production of gray-scale raw images, susceptibility to technical artifacts, and reliance on the expertise of the reader for accurate interpretation [93]. AI can help to overcome these limitations. Algorithms used in melanoma diagnosis with RCM are summarized in Table 4.

Due to the nature of the technique, RCM is susceptible to various artifacts that may impact image quality and diagnostic accuracy. These include faulty reflectance caused by corneal layer reflection or foreign objects such as air or oil bubbles, artifacts from the convexity of nodular lesions or skin creases, and motion artifacts like shifting and misalignment of RCM mosaics due to subtle movements by the patient or technician. Different AI techniques can be employed to detect and eliminate these artifacts. Kose et al. showed that an automated semantic segmentation method called Multiscale Encoder–Decoder Network (MED-Net) could automatically detect artifacts in RCM images of melanocytic lesions with 83% sensitivity and 92% specificity [94].

Another pitfall of RCM is the significant training required for accurate image interpretation, which is essential for achieving high diagnostic accuracy. Gerger et al. developed an automated diagnostic image analysis system using Classification and Regression Trees (CARTs) to differentiate melanoma from benign nevi in RCM images. The system correctly classified 97.31% of the images in the learning set and 81.03% in the test set [95]. Koller et al. employed a similar machine learning algorithm using Classification and Regression Trees (CART) analysis software to distinguish between benign melanocytic nevi and melanoma in RCM images [96]. The algorithm successfully classified 93.60% of the melanoma images and 90.40% of the nevi images within the learning set. However, its success did not extend to an independent test set, indicating limitations in its generalizability. Wodzinski et al. used a CNN based on ResNet architecture, which achieved an 87% accuracy rate in identifying common skin neoplasms, such as melanoma, BCC, and nevi, using in vivo RCM images. This performance slightly surpassed the diagnostic accuracy of human experts [97]. Kose et al. developed an automated semantic segmentation method known as the Multiscale Encoder–Decoder Network (MED-Net). MED-Net was tested on an international dataset selected to reflect the data diversity encountered in daily clinical practice. Their findings demonstrated that MED-Net with the “deep supervision” method achieved a pixel-wise mean sensitivity of 70 ± 11% and a specificity of 95 ± 2% for detecting various patterns of melanocytic lesions at the dermal/epidermal junction (DEJ) in in vivo RCM images. Furthermore, MED-Net accurately identified the location and extent of these patterns, achieving a Dice coefficient of 0.71 ± 0.09 [98].

Similarly, D’Alonzo et al. implemented a weakly supervised semantic segmentation model based on EfficientNet, a deep neural network (DNN), to analyze RCM mosaics of pigmented lesions at the dermal–epidermal junction (DEJ). This model was designed to distinguish between non-worrisome (“benign”) areas and those suggestive of melanoma (“aspecific”). The trained model achieved an average area under the ROC curve of 0.969 and a Dice coefficient of 0.778, demonstrating the potential for spatial localization of aspecific regions in RCM images, thereby enhancing the interpretability of diagnostic decisions for clinicians [99]. Finally, Mandal et al. aimed to distinguish Lentigo maligna (LM) from atypical intraepidermal melanocytic proliferation (AIMP). The authors developed a method that first merges an RCM stack into a single image via local-z projection [100] and then processes the resulting image using DenseNet169, a CNN classifier. It was trained and tested over a dataset of 517 RCM stacks (389 LM and 148 AIMP) collected from 110 patients (split into ~80% training vs. ~20% testing). The model achieved an accuracy of 0.80 [101].

**Table 4 life-14-01602-t004:** Algorithms used in melanoma diagnosis with RCM.

Publication	End-Point	Dataset	Algorithm	Performance
Kose et al. [94]	Segmentation; detection of artifacts	117 RCM mosaics; obtained from 7 clinics that collaborated internationally, the ethnicity and skin types of patients were not specified	MED-Net; an automated semantic segmentation method	Sensitivity: 82%, Specificity: 93%
Gerger et al. [95]	Classification; benign nevi vs. melanoma	408 benign nevi and 449 melanoma images; obtained from one clinic in Austria, the ethnicity and skin types of patients were not specified	CART (Classification and Regression Trees)	Learning set: 97.31% of images correctly classified Training set: 81.03% of images correctly classified
Koller et al. [96]	Classification; benign nevi vs. melanoma	4669 melanoma and 11,600 benign nevi RCM images; obtained from one clinic in Austria, the ethnicity and skin types of patients were not specified	CART (Classification and Regression Trees)	Learning set: 93.60% of the melanoma and 90.40% of the nevi images were correctly classified
Wodzinski et al. [97]	Classification; benign nevi vs. melanoma vs. BCC	429 RCM mosaics; obtained through collaboration with two clinics from Italy and Poland, the ethnicity and skin types of patients were not specified	a CNN based on ResNet architecture	F1 score for melanoma in test set: 0.84 ± 0.03
Kose et al. [98]	Segmentation; six distinct patterns (aspecific, non-lesion, artifact, ring, nested, meshwork)	117 RCM mosaics; acquired at 4 different clinics in the US and a clinic in Italy, the ethnicity and skin types of patients were not specified	an automated semantic segmentation method, MED-Net	Pixel-wise mean sensitivity: 70 ± 11% Pixel-wise mean specificity: 95 ± 2%, respectively, with 0.71 ± 0.09 Dice coefficient over six classes.
D’Alonzo et al. [99]	Segmentation; “benign” and “aspecific (nonspecific)” regions	157 RCM mosaics; obtained from 4 different clinics in the US and a clinic in Italy, the ethnicity and skin types of patients were not specified	Efficientnet, a deep neural network (DNN)	AUC of 0.969, and Dice coefficient of 0.778
Mandal et al. [101]	Classification; Atypical intraepidermal melanocytic proliferation (AIMP) vs. Lentigo Maligna (LM)	517 RCM stacks (389 LM and 148 AIMP) from 110 patients attended two clinics in Australia, the ethnicity and skin types of patients were not specified	DenseNet169, a CNN classifier	Accuracy: 0.80 F1 score for LM: 0.87

Reflectance confocal microscopy (RCM); CART (Classification and Regression Trees); Basal Cell Carcinoma (BCC); Convolutional neural network (CNN); Deep neural network (DNN); Atypical intraepidermal melanocytic proliferation (AIMP); Lentigo Maligna (LM).

#### 3.3.2. Optical Coherence Tomography (OCT) and OCT-like Devices

Optical Coherence Tomography (OCT) is a non-invasive imaging method that captures the echo delays and intensity of reflected infrared or near-infrared light [102]. It enables real-time visualization of the skin, with the ability to penetrate depths of 1–2 mm and deliver a resolution ranging from 3 to 15 µm [103]. Building on the principles of OCT technology, several new devices, such as full-field OCT (FF-OCT), vibrational optical coherence tomography (VOCT), and combination devices like an OCT module with near-infrared Raman spectroscopy, have been developed to enhance accuracy and image quality.

AI has been specifically utilized to assist with image interpretation at various levels in these devices. The delineation of the dermal–epidermal junction (DEJ) is also crucial for diagnosing melanoma in OCT images. Chou et al. utilized a multi-directional CNN to successfully predict the DEJ in FF-OCT images [104]. Silver et al. demonstrated that a machine learning model based on logistic regression achieved a specificity of 77.8% and a sensitivity of 83.3% in distinguishing melanoma from normal skin in VOCT images [105]. Lee et al. trained an SVM using OCT images of melanoma and benign nevi and subsequently applied this machine learning model to successfully identify pigmented non-malignant lesions in a patient with phacomatosis pigmentokeratotica [106]. You et al. developed an integrated OCT-Raman spectroscopy device and utilized several machine learning models to differentiate between various skin cancer cell types (BCC, SCC, and melanoma) and normal cells in experimentally cultivated cell line models. By applying the decision tree algorithm to OCT features, an accuracy of 85.9% was obtained in distinguishing between cancerous and healthy cells. Moreover, impressively, the discrimination accuracy between melanoma and keratinocytic tumors using all Raman spectra reached 98.9% with the KNN algorithm and 91.6% with the decision tree (TREE) algorithm [107].

## 4. Comparison of AI to Traditional Methods

The use of AI in the detection of skin pathologies, such as melanoma and non-melanoma skin cancers, appears promising in identifying suspicious lesions and may complement more traditional measures such as biopsies. Biopsies offer a histopathological diagnosis with high specificity and sensitivity; however, they are labor-intensive, invasive, and may expose patients to infection. These drawbacks can delay diagnosis, particularly in regions with limited dermatopathology resources.

In contrast, AI-based image analyses and non-invasive imaging, such as dermoscopy and confocal microscopy, provide rapid preliminary assessments that are potentially more accessible for regular screenings, especially in regions lacking sufficient resources or in telemedicine settings. Artificial intelligence can enable large-scale screening by automating lesion classifications and triage, helping streamline lesions that warrant further investigation. These approaches could reduce the frequency of unnecessary biopsies by enhancing diagnostic accuracy at the point of initial assessment, especially in underserved settings or where access to dermatologists is limited.

Prior literature has reported that AI-based algorithms can yield high sensitivity and specificity (87% and 77.1%, respectively) in skin cancer diagnosis, which is comparable to or exceeds the clinician’s performance for certain lesion subtypes [108]. AI models have outperformed clinicians with less experience in binary classification tasks, such as determining whether a lesion is benign or malignant, implicating the potential of AI to improve diagnostic accuracy among less experienced practitioners or in resource-limited settings.

Notably, despite the significant improvement of AI, these algorithms cannot fully replace biopsy for definitive melanoma diagnoses, especially among patients with di-verse lesion types and in patients of skin of color. Nonetheless, a combination of AI-based lesion triage and follow-up of low-risk lesions (e.g., non-melanoma skin cancers) will potentially start to take place in the clinical setting, reducing the need for biopsies substantially. Future advancements, combined with improved image quality and training datasets that capture a broader diversity of skin types and colors, and lesion variations could make these tools even more valuable as adjuncts to biopsy in melanoma screening and management.

## 5. Limitations

### 5.1. Datasets

#### 5.1.1. Skin Type Diversity

There are imbalances in terms of ethnicity and skin tone in the datasets [109]. Most publicly available skin image datasets predominantly consist of images of white/fair-skinned individuals or lack labels for skin type [110,111]. A minority of studies include images of darker skin types to train or test AI algorithms for skin cancer diagnosis, indicating potential shortcomings in using various datasets [112]. State-of-the-art algorithms trained on datasets primarily composed of white/fair-skinned skin types have been tested on a dataset that includes diverse skin types. However, when applied to lesions from individuals with darker skin tones, the performance of these algorithms declines significantly compared to their performance on the demographics present in their training data [113,114]. Similarly, an algorithm trained mainly on East Asian skin images performed poorly on skin lesion images of White American patients [111]. These findings indicate the significance of the distribution of skin types within datasets. Moreover, datasets should include Fitzpatrick skin type metadata to evaluate skin type diversity properly and appropriately train models [32]. When datasets were evaluated based on skin type metadata, only 2.1% of images included skin type labels [111]. In the Fitzpatrick 17k dataset, researchers manually added skin type labels to existing atlas images [32,114]. Nevertheless, given the infrequency of datasets containing skin type labels, it remains necessary to develop datasets that address this issue [109].

#### 5.1.2. Metadata

The majority of current datasets have deficiencies in comprehensive metadata [110,111]. Nonetheless, to enhance algorithm performance and facilitate generalization, datasets should incorporate extensive metadata, including demographic and clinical details alongside image acquisition methodologies. It is particularly important to integrate information such as age, gender, ethnicity, lesion anatomical localization, genetic influences, environmental factors, and socioeconomic status within datasets, as these factors play a crucial role in clinical diagnosis and monitoring. For example, individuals with fair skin, numerous moles, and a family history of melanoma are at a higher risk of developing skin cancer. Furthermore, the likelihood of developing skin cancer significantly rises when environmental risk factors, like prolonged UV light exposure, are combined with these factors. The inclusion of such factors can enhance deep learning techniques [112,113,114]. In the ISIC 2019 challenge, the dataset included metadata such as age, gender, and anatomical location, resulting in improved average algorithmic accuracy [115]. Likewise, Haenssle et al. demonstrated an increase in sensitivity in dermoscopy management decisions and sensitivity and specificity in diagnostic performance through the integration of clinical information [32,116].

#### 5.1.3. Combination of Different Modalities

Image datasets predominantly contain dermatoscopic images followed by macroscopic clinical photographs. Few datasets include matched images from multiple modalities [111]. Furthermore, to date, published datasets incorporating confocal microscopy or total body photography (TBP) images are lacking [117,118]. Datasets that match versions of the same images obtained through different modalities enhance algorithm utility and success in clinical practice, as they create an environment more akin to real-world applications [42]. When combined with images from total body photography and lesion follow-ups, these datasets can assist in automatic mole mapping. This approach may facilitate the development of a screening method for distinguishing between lesions that do not require detailed evaluation, similar to clinical assessments, and those that necessitate further evaluation. Additionally, it could aid in the early detection of potential skin cancers that may arise during patient follow-ups [117,118].

#### 5.1.4. Rare Subtypes

Generally, there has been limited representation of rare skin cancers in datasets. Nonetheless, these rare cancers may be critical to disregard due to their aggressive clinical behavior [119]. For instance, while algorithms may effectively detect melanoma, this type of cancer encompasses various subtypes with distinct visual features and prognostic differences [120,121]. The majority of cancer images in datasets predominantly feature melanoma, followed by more common keratinocyte skin cancers such as basal cell carcinoma and squamous cell carcinoma [111]. Studies have indicated weaker algorithm performance in melanoma subtypes like amelanotic melanoma and subungual melanoma [81,122]. Highlighting this issue is crucial, and promoting the reporting of subgroup performance for rarer skin cancers in studies is essential for determining the optimal performance of algorithms [114].

#### 5.1.5. Diagnostic Method

The diagnostic methods for some lesions are not specified in certain datasets [114]. When lesions are incorrectly diagnosed using clinical diagnoses during the training of algorithms, the likelihood of the algorithm making errors increases [123]. Hekler et al. conducted a study comparing the performance of models trained on datasets labeled by the majority opinion of dermatologists with those trained on datasets labeled with histopathological diagnoses. The model utilizing clinical diagnoses achieved an accuracy of 64.2%, whereas the model using histopathological diagnoses reached an accuracy of 73.8% [124]. This highlights the importance of datasets comprised of lesions diagnosed using the gold standard method. Nevertheless, despite the enhanced accuracy associated with histopathology, performing a biopsy on every lesion is challenging due to its invasive nature. This challenge is compounded when considering the collection of training sets involving thousands of lesions, potentially leading to ethical issues [125]. Furthermore, even in histopathological diagnosis, there may be instances where histopathologists do not reach a consensus [126]. Therefore, to facilitate accurate comparisons between algorithms, it is essential to detail the diagnostic methods used in the datasets [127,128].

#### 5.1.6. Image Quality

Another limitation of the datasets are the differing camera hardware, zoom level, focus, lighting, field of view, and presence of artifacts of each of the datasets, which impact how the different ML models can predict/classify lesions. As demonstrated by Mier et al., differing degrees of blurring and brightness can negatively impact the efficacy of ML models, reducing accuracy and sensitivity [129]. Wrinkler et al. compared the results of a machine learning model when using images with surgical skin markings to images that do not involve such markings. They found that the specificity and AUC dropped, indicating that artifacts like surgical skin markings can have a drastic impact on a model’s performance [130]. The same group evaluated the effect of the dark corner artifact (DCA) found in dermoscopic images on ML performance and found that their model had comparable results between no DCA and small DCA and medium DCA, but a significant decrease in specificity with an unchanged AUC when comparing no DCA to large DCA [131].

ML utilizes patterns and features in images rather than adapting to the differences in imaging conditions (such as brightness, artifacts, and resolution), subjecting the datasets that have these variations to potentially lower AUCs/predicting capabilities. This could also affect the model’s generalizability as there could be potential differences in the resolution of the images in the datasets and those of images used in clinical practice.

### 5.2. Generalizability

One of the primary challenges in using artificial intelligence for melanoma diagnosis is ensuring the generalizability of machine learning models across different patient cohorts. A comprehensive meta-analysis has shown that automated systems generally perform worse in studies using independent test sets than in those with non-independent test sets for computer-aided melanoma diagnosis [132]. This issue largely stems from fundamental aspects of AI workflows, particularly the quality of the datasets on which machine learning models are trained, which has a substantial impact on their performance with independent test data [133]. For example, if the training dataset is small or composed predominantly of complex, outlier features specific to a limited group of patients, the model may fail to learn realistic associations between features and produce poor results in independent test sets, a phenomenon known as overfitting [134]. Factors such as dataset size, diversity, balanced sampling, and the number of variables in the dataset can all affect training quality, making it challenging to achieve ideal generalizability. More specifically, machine learning models tend to perform well under conditions similar to those in which they were trained. For example, if a model is trained on data from a tertiary clinic, it is likely to perform best in similar clinical settings. Likewise, a model trained on a population with predominantly fair skin types likely performs well within that demographic but may be less accurate for populations with different skin types [135]. Consistent with these inferences, our review found few studies that thoroughly tested generalizability on globally representative “benchmark” datasets. Moreover, we observed that most studies did not specify essential clinical and demographic characteristics of the patients in their datasets, such as skin type, ethnicity, or geographic location, raising concerns about the generalizability of their findings.

Another significant issue affecting model generalizability in melanoma diagnosis is class imbalance, where instances of one class (e.g., benign nevi) far outnumber those of other classes (e.g., melanoma) in the datasets. This can lead to training models that are biased towards the more frequent “majority” class, favoring it over minority classes unless special attention is paid through class balancing techniques such as resampling the minority class, downsampling the majority class, or class re-weighting [136].

Our review revealed significant variability in the distribution of skin lesion types across studies, with different counter-measures applied to address class imbalance. Furthermore, certain rare or diagnostically complex entities, such as atypical or dysplastic nevi, amelanotic melanoma, and acral lentiginous melanoma, were underrepresented in most of the reviewed studies. These findings raise important questions about the generalizability of models trained on such datasets.

Moreover, our analysis suggests that most existing datasets do not accurately reflect class imbalance, as they are primarily collected at specialized care centers (e.g., cancer care clinics) with higher malignancy rates compared to primary care clinics. Therefore, ongoing initiatives should focus on developing publicly available, large-scale “benchmark” datasets to facilitate the evaluation of AI model generalizability in melanoma diagnosis [137]. To address these challenges, dataset development must prioritize reflecting the patient population and demographics at different levels of patient care.

### 5.3. Reliability

Understanding how certain machine learning models operate—particularly black-box models like CNNs, DNNs, and transformers—is inherently challenging for the human brain, and thus for clinicians. Therefore, various methods have been devised to explain how machine learning models make decisions, aiming to improve the reliability of AI methods. These explainable AI methods and natural language processing models are also effective in increasing the clinician’s confidence. To this end, Kim et al. developed a vision-language model called MONET [138] that is trained using figure and caption pairs from dermatology literature. The model can correlate the dermatological concepts in the captions with the information in the images in a self-supervised fashion. The resulting model can accurately annotate concepts in dermatology images, as verified by dermatologists, surpassing the performance of supervised models built on previously annotated datasets of clinical images. In this way, MONET provides AI transparency and interpretability throughout the development pipeline, from designing inherently interpretable models to datasets and model auditing. Yan et al. proposed an Explanatory Interactive Learning method that integrates human users into the training process of machine learning for diagnosing skin malignancies. This approach helps identify and remove confounding behaviors of ML models by transforming their feature representations into explainable concept scores for human users. Additionally, this method improves diagnostic accuracy for melanoma by addressing confounding factors, such as air pockets and dark corners [139].

As machine learning models continue to improve and are made publicly available through open-source code and model-sharing initiatives (e.g., winner models from competitions like the ISIC ML challenge being shared via the ISIC code repository), we enter a new phase of adoption. The next significant step is evaluation of these models in real-world clinical settings, where prospective trials will assess their reliability across diverse populations and clinical scenarios [140,141]. This phase will provide clinicians with a deeper understanding of these models’ potential value and limitations within their own practice, ultimately informing evidence-based decision-making.

## 6. Conclusions

As dermatologists inherently rely on visual assessments, it remains a field well-suited for the integration of artificial intelligence. The increasing incorporation of AI into dermatological practice holds promise in mitigating clinician workload by optimizing the identification and prioritization of benign lesions at primary care settings, thereby reducing unnecessary invasive testing and ultimately diminishing morbidity and mortality rates. However, despite these advancements, achieving a 100% accurate diagnosis with AI remains an elusive goal, occasionally leading to overdiagnosis and overtreatment of malignant lesions, particularly premalignant lesions that may not progress to cancer.

Another significant challenge arises from the generalizability of AI models to diverse patient populations not adequately represented in training datasets. This issue can manifest in various ways, including differences in image acquisition devices, methods, and demographic characteristics between datasets. Furthermore, the existing literature highlights the need for increased diversity in dermatological imaging datasets, as they predominantly feature individuals with lighter skin tones. This underrepresentation underscores the importance of developing and utilizing AI models that incorporate diverse patient populations to minimize performance disparities.

To address the challenges of generalizability and diversity in AI-driven skin cancer diagnosis, large-scale, inclusive datasets are being developed as standardized benchmarks for evaluating AI performance. The International Skin Imaging Collaboration (ISIC) has played a major role in this effort by aggregating over 1.1 million images from leading cancer research institutions on five continents. Through its regular machine learning challenges, ISIC fosters collaboration between the AI and medical communities, enabling researchers to compare their approaches against standardized datasets and accelerate the development of more accurate and effective AI models for skin cancer diagnosis. Furthermore, ISIC leads the efforts within Digital Imaging and Communications in Medicine (DICOM) [142] for dermatology imaging modalities, driving the standardization of image acquisition protocols and metadata collection practices to minimize discrepancies between datasets. This collaborative approach enables the creation of a more robust and representative dataset, which is essential for developing AI models that can accurately diagnose skin cancers in diverse patient populations. By promoting interoperability and consistency across datasets, ISIC facilitates the development of more reliable and generalizable AI solutions for skin cancer diagnosis.

Despite these advancements, it is essential to acknowledge that dermatological expertise and clinical correlation remain indispensable for ensuring precision in diagnostic evaluations and treatment strategies, particularly for complex cases requiring contextual insights.

## Figures and Tables

**Figure 1 life-14-01602-f001:**
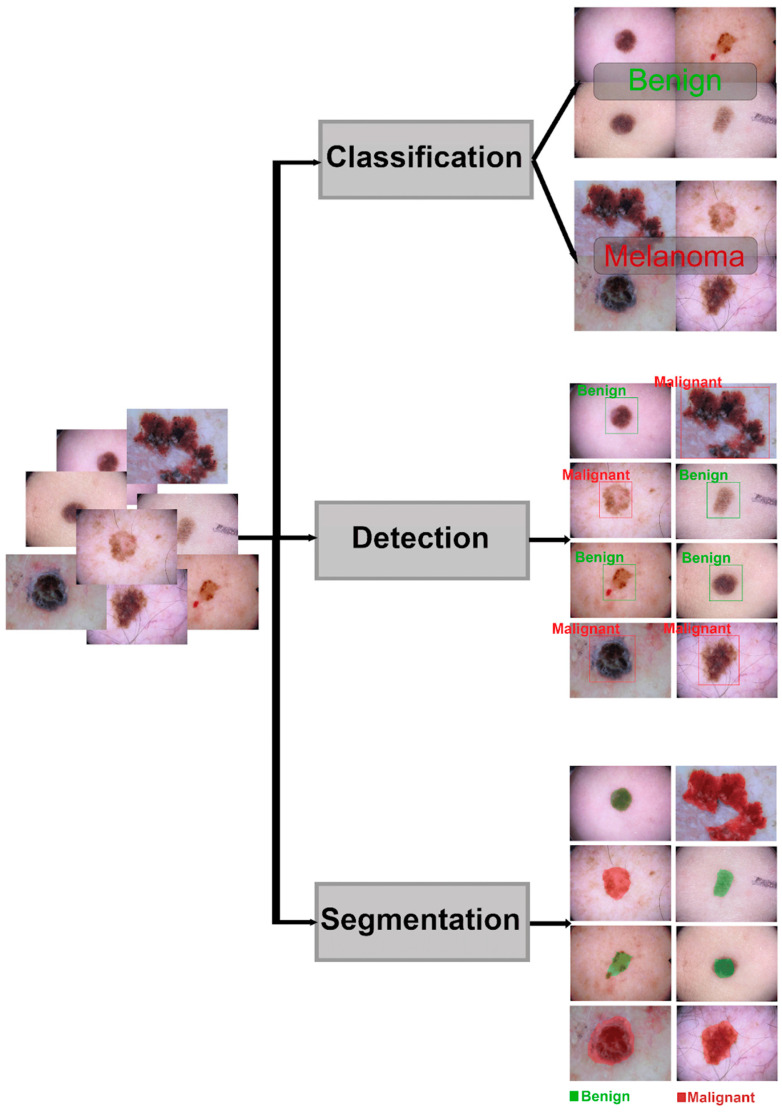
Classification, detection, and segmentation tasks in dermatology image analysis. Images and their label information are taken from the ISIC Archive. The segmentations are for illustrative purposes only and may not show the exact borders of the lesions.

**Table 1 life-14-01602-t001:** Algorithms used in the diagnosis of melanoma from clinical images.

Publication	End-Point	Dataset	Algorithm	Performance
Nasr-Esfahani et al. [41]	Classification (benign/melanoma)	170 clinical images that underwent data augmentation to generate 6120 images (80% training, 20% validation). Ethnicity was not specified.	CNN with 2 convolutional layers each followed by pooling layers along with a fully connected layer	Acc: 81% Spe: 80% Sen: 81% NPV: 86% PPV: 86%
Yap et al. [42]	Classification of melanoma from 5 different types of lesions	2917 cases with each case containing patient metadata, macroscopic image and dermoscopic images with 5 classes (naevus, melanoma, BCC, SCC, and pigmented benign keratoses). Not specified where images were obtained from or ethnicity of images.	ResNet-50 with embedding networks	**Macroscopic images alone** AUC: 0.791 **Macroscopic and dermoscopy** AUC: 0.866 **Macroscopic, dermoscopy and metadata** AUC: 0.861
Riazi Esfahani et al. [43]	Classification (malignant melanoma/benign nevi)	793 images (437 malignant melanoma and 357 benign nevi). Ethnicity not specified.	CNN	Acc: 88.6% Spe: 88.6% Sen: 81.8%
Dorj et al. [44]	Classification of melanoma from 4 different skin cancers (actinic keratoses, BCC, SCC, melanoma)	3753 images (2985 training and 758 testing) including 958 melanoma. Ethnicity not specified.	AlexNet with ECOC-SVM classifier	Acc: 0.942 Spe: 0.9074 Sen: 0.9783
Soenksen et al. [45]	Classification across 6 different classes as well as distinguishing SPLs	33,980 (including backgrounds, skin edges, bare skin sections, low priority NSPLs, medium priority NSPLs and SPLs) (60% training, 20% validation and 20% as testing). Ethnicity not specified.	DCNN with VGG16 Image Net pretrained network as transfer learning	**Across all 6 classes** AUC_micro_: 0.97 Spe_micro_: 0.903 Sen_micro_: 0.899 **For SPLs** AUC: 0.935
Pomponiu et al. [46]	Classification (melanoma/benign nevi)	399 images (217 benign, 182 melanoma) from online image libraries. Ethnicity not specified.	CNN with a KNN classifier	Acc: 0.83 Spe: 0.95 Sen: 0.92
Han et al. [30]	Melanoma detection from 12 different skin diseases	Training: 19,938 images from the Asan dataset [29], MED-NODE dataset [31], and atlas site images. Testing: 480 images from Asan and Edinburgh datasets [25]. Asan dataset was composed of mainly an Asian population and Edinburgh and MED-NODE were mainly composed of a Caucasian population.	ResNet152	**Asan** AUC: 0.96 Spe: 0.904 Sen: 0.91 **Edinburgh** AUC: 0.88 Spe: 0.855 Sen: 0.807
Liu et al. [47]	Primary: classification among 26 different skin conditions Secondary: classification among a full set of 419 different skin conditions	Training: 64,837 images with metadata. Validation set A: 14,833 images with metadata. Validation set B was used to compare to dermatologists: 3707 images with metadata. Training: 0.1% American Indian or Alaska Native, 11% Asian, 6.8% African American, 43.7% Hispanic, 1.4% Native Hawaiian/Pacific Islander, 34% White, 2.2% not specified. Validation A: 0.1% American Indian or Alaska Native, 12.6% Asian, 6.1% African American, 43.4% Hispanic, 1.6% Native Hawaiian/Pacific Islander, 31.3% White, 3.9% not specified Validation B: 0.9% American Indian or Alaska Native, 10.1% Asian, 6.3% African American, 42.5% Hispanic, 2% Native Hawaiian/Pacific Islander, 34.2% White, 4% not specified.	DLS with Inception-v4 modules and shallow module	**Validation set A for 26 image classification:** Acc_top1_: 0.71 Acc_top3_: 0.93 Sen_top1_: 0.58 Sen_top3_**:** 0.83 **Validation set B for 26 image classification:** Acc_top1_: 0.66 Acc_top3_: 0.9 Sen_top1_: 0.56 Sen_top3_: 0.64 **Dermatologists:** Acc_top1_: 0.63 Acc_top3_: 0.75 Sen_top1_: 0.51 Sen_top3_: 0.49
Sangers et al. [48]	Classification (low/high risk)	785 images (418 suspicious, 367 benign). Ethnicity not specified.	RD-174	**Overall app classification** Sen: 0.869 Spe: 0.704 **Classification for melanocytic lesions:** Sen: 0.819 Spe: 0.733
Polturu et al. [49]	Classification (non-melanoma/melanoma)	206 images from DermIS [28] and Derm Quest [27] (87 nonmelanoma and 119 melanoma, 85% used for training and 15% used for testing). Ethnicity not specified.	AutoML was created using a no-code online service platform	Acc: 0.844 Sen: 0.833 Spe: 0.857

Convolutional Neural Network (CNN); Accuracy (Acc); Specificity (Spe); Sensitivity (Sen); Negative Predictive Value (NPV); Positive Predictive Value (PPV); Basal Cell Carcinoma (BCC); Squamous Cell Carcinoma (SCC); Area under the ROC curve (AUC); Error-Correcting Output Codes (ECOC); Support Vector Machine (SVM); Suspicious Pigmented Lesions (SPLs); Nonsuspicious Pigmented Lesions (NSLPs); Deep Convolutional Neural Network (DCNN); Visual Geometry Group (VGG); k nearest neighbor (KNN); Deep learning system (DLS); Automated machine learning (AutoML).

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
