# Peer review of "Artificial Intelligence in the Non-Invasive Detection of Melanoma"

_life, 2024, doi:10.3390/life14121602_

Round 1
Reviewer 1 Report
Comments and Suggestions for Authors
The manuscript is a comprehensive review of the use of AI models in melanoma diagnostics. I've found some issues that have to be corrected. I strongly believe that resolving those issues will improve the scientific soundness of the manuscript.
Major:
1) The diversity of datatests have to be more specifically discussed in the text, eg. lines 148-149. Please elaborate on the ethnicity, skin types and geographic regions with regard to data used in mentioned datasets
2) Although the paper discusses the potential of AI use in diagnostics, it feels like the discussion would be much strengthen if these methods were compared with traditional diagnostic methods like biopsies
3) Either in the introductory or discussion sections, the differences between DL, CNN and transformers should be clearly described to make them easier to understand with less advanced readers
3) The manuscript lacks critical analysis of the limitations of AI models in practical approaches including datasets limitations, variable image quality, user trust and model validation in different settings.
Minor:
1) l.15-16 - the phrase "much less common" followed by "is the most deadly" feels redundant. I suggest rephrasing the sentence to something like: "Although melanoma is less common than basal or squamous cell carcinomas, is the deadliest form of cancer"
2) Sometimes the singular verb "has" is used when referring to multiple subjects. Please double-check manuscript for that instance.
Author Response
Please find the response to reviewer file below. Thank you

Reviewer 2 Report
Comments and Suggestions for Authors
This is an impressive and easy to read review of mostly melanoma artificial intelligent diagnostic tools. The authors have synthesized and enormous amount of information. My concerns are minor.
1. Throughout the manuscript, the authors either use commas or periods to indicate decimal points. This notation should be consistent.
2. In table 3, I do not understand the meaning of the characters "Melanoma: 0,0,94" Similarly the notation "DenseNet 201: 93,80" is unclear.
3. A high-risk individual being screened for melanoma may often have 100 or more melanocytic nevi and 0 or at most 1 melanoma. I am not a melanoma expert, but in a typical clinic day I will see in excess of 1000 pigmented spots raning from benign nevi, dermatofibromas to a few various cancers. It is not realistic to examine all of these dermoscopically nor pathologically. In the manuscript the models discussed were trained with samples with much higher proportions of cancers. Moreover, many high-risk patients bear atypical or dysplastic nevi, with severities from mild to severe. I argue the detection of a severely dysplastic nevus is important for a useful AI system. Aside from the notion that there are no universally accepted pathologic criteria for the diagnoses of atypical nevi, these unusual melanocytic nevi are important to accurately diagnose and distinguish from melanomas. For these tools to be useful, much larger sample sizes of benign melanocytic nevi, seborrheic keratoses, and other benign tumors as well as atypical nevi, must be used to train and test the systems. The authors should work a bit harder to discuss these major limitations.
Author Response
Please find response to Reviewers' file below. Thank you.

Reviewer 3 Report
Comments and Suggestions for Authors
I have reviewed the manuscript: “Artificial Intelligence in the Non-Invasive Detection of Melanoma”. I have the following comments:
- The manuscript is well written with a good overall review of the subject. The English language is understandable and fluent.
- The title is appropriate for the content of the manuscript. Each section provides appropriate review and updated information for the readers. All four tables are robust and add to the content of the manuscript.
- The conclusion appropriately sums up all the pertinent information. The manuscript is well-referenced.
- One shortcoming I noticed is the lack of images/figures. If any pertinent images are available, they can be added to improve the overall quality of the manuscript.
Thank you.
Author Response

(The authors gave the same response as above.)

Round 2
Reviewer 1 Report
Comments and Suggestions for Authors
Dear Authors,
I am now satisfied with the manuscript in its current version.
Congratulations!
Author Response
Dear Reviewer,
Thank you for your valuable feedback, and for helping us in enhancing our article.